# DECEPTIVE-NERF:
# ENHANCING NERF RECONSTRUCTION USING PSEUDO-OBSERVATIONS FROM DIFFUSION MODELS

## ABSTRACT

We introduce *Deceptive-NeRF*, a novel methodology for few-shot NeRF reconstruction, which leverages diffusion models to synthesize plausible *pseudo-observations* to improve the reconstruction. This approach unfolds through three key steps: 1) reconstructing a coarse NeRF from sparse input data; 2) utilizing the coarse NeRF to render images and subsequently generating pseudo-observations based on them; 3) training a refined NeRF model utilizing input images augmented with pseudo-observations. We develop a *deceptive diffusion model* that adeptly transitions RGB images and depth maps from coarse NeRFs into photo-realistic pseudo-observations, all while preserving scene semantics for reconstruction. Furthermore, we propose a progressive strategy for training the Deceptive-NeRF, using the current NeRF renderings to create pseudo-observations that enhance the next iteration's NeRF. Extensive experiments demonstrate that our approach is capable of synthesizing photo-realistic novel views, even for highly complex scenes with very sparse inputs. Codes will be released.

## 1 INTRODUCTION

Since its debut Neural Radiance Fields (NeRFs) (Mildenhall et al., 2020) have achieved unprecedented results in novel view synthesis to date. While producing visually pleasing results, a vanilla NeRF requires a large number of training views and is prone to generating severe artifacts when dealing with particularly sparse observations. This issue considerably hampers the further and more practical applications of NeRFs, considering the casual data collection conditions of lay users, such as one where images are collected using their mobile devices.

To address this issue, recent works have explored several strategies. Pre-training approaches leverage large-scale datasets comprising various scenes for injecting prior knowledge (Yu et al., 2021b; Chen et al., 2021; Chibane et al., 2021; Jang & Agapito, 2021; Johari et al., 2022). Regularization approaches employ a range of regularizations derived from depth supervision, patch rendering, semantic consistency, visibility, or frequency pattern (Deng et al., 2022b; Roessle et al., 2022; Guangcong et al., 2023; Niemeyer et al., 2022; Jain et al., 2021; Seo et al., 2023b; Wynn & Turmukhambetov, 2023; Somraj & Soundararajan, 2023; Yang et al., 2023; Seo et al., 2023a). Although these techniques have contributed in improving the reconstruction quality of few-shot NeRF, undesirable artifacts can still be observed in the synthesized novel views, where tailored heuristic factors specific to individual scenes are still needed to generate usable results.

Recent progress in image synthesis using diffusion models (Ho et al., 2020; Sohl-Dickstein et al., 2015; Rombach et al., 2022; Zhang & Agrawala, 2023) boosts 3D content generation, by transferring the natural image prior learned from Internet-scale 2D data to 3D settings (Deng et al., 2022a; Xu et al., 2022; Melas-Kyriazi et al., 2023; Zhou & Tulsiani, 2022; Liu et al., 2023; Chan et al., 2023; Gu et al., 2023) [1]. An intuitive approach to utilizing diffusion models for few-shot novel view synthesis is to employ them as a "scorer" to evaluate the quality of NeRF-rendered images and thus a regularizer for NeRF training. This approach however necessitates a large diffusion model be inferred at each training step of the radiance field, which is a very computationally intensive process.

---

[1]Note 3D content generation from images differs fundamentally from few-shot novel view synthesis. This work tackles the latter where the goal is "reconstruction" rather than "generation".

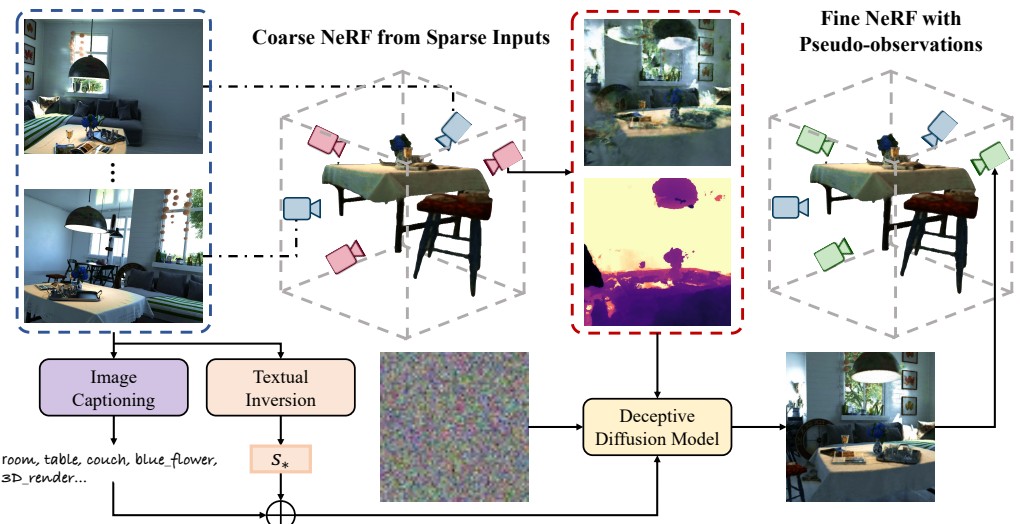

Figure 1: **Overview of Deceptive-NeRF.** 1) Given a sparse set of □ input images associated with their camera poses, we first train a coarse NeRF to render □ coarse novel view images and depth maps. 2) We use a deceptive diffusion model to fine-tune RGB-D images from the coarse NeRF to synthesize □ pseudo-observations from corresponding viewpoints. 3) We train a fine NeRF using both input images (real) and pseudo-observations (fake) as our final reconstruction of the scene while enforcing consistency across the fake images from different viewpoints.

In this paper, we propose *Deceptive-NeRF*, a strategy that efficiently leverages large diffusion models for few-shot NeRF reconstruction, as shown in Figure 1. Instead of using diffusion models only as a means to regularize the quality of NeRF-rendered images, we directly take the images produced by diffusion models as auxiliary observations, complementing the sparse inputs, to train a NeRF. Specifically, our method consists of three key steps: 1) reconstruct a coarse NeRF model from given sparse views; 2) generate *pseudo-observations* based on the coarse model renderings; 3) train a fine NeRF model from both input views and pseudo-observations to produce a high-quality reconstruction. To generate plausible pseudo-observations consistent with the input views, we propose a *deceptive diffusion model*, refining coarse RGB and depth images. This novel approach tackles the issue of sparsity by "densifying" observations, while not demanding excessive time or computation, thanks to the one-time usage of diffusion models. We further propose a progressive training strategy that at each iteration uses the current NeRF model renderings to generate pseudo-observations for the training of the next iteration's NeRF. In summary, our contributions include the following:

- We propose a novel approach for few-shot novel view synthesis that leverages large diffusion models to generate pseudo-observations, instead of using them as a "scorer" to provide training signals.

- To generate photo-realistic pseudo-observations that faithfully preserve scene semantics and input view consistency, we propose a deceptive diffusion model.

- Extensive experiments and ablation studies validate our key design choices and demonstrate improvements over current state-of-the-art methods for few-shot novel view synthesis.

## 2 RELATED WORK

**Novel view synthesis via NeRF.** Novel view synthesis, the problem of synthesizing new viewpoints given a set of 2D images, has recently attracted much attention. Using continuous 3D fields and volumetric rendering, Neural Radiance Fields (NeRFs) (Mildenhall et al., 2020) have enabled a new and effective approach for novel view synthesis. Follow-up works have since emerged to enhance NeRFs and expand their applications, such as modeling dynamic scenes (Zhang et al., 2021; Park et al., 2021; Pumarola et al., 2021; Tretschk et al., 2021), acceleration (Yu et al., 2021a; Fridovich-Keil et al., 2022; Chen et al., 2022; Müller et al., 2022), and 3D scene editing (Liu et al., 2021; Zhang et al., 2021; Wang et al., 2022; Jang & Agapito, 2021; Kobayashi et al., 2022). Despite significant progress, NeRFs still struggle to synthesize novel views when there are only a limited number of input views, i.e., when handling few-shot novel view synthesis.

**Few-shot NeRF.** Several studies have been conducted to enhance the rendering quality of NeRF when provided with only sparse observations. Pre-training methods (or transfer learning techniques) utilize prior knowledge from extensive datasets of 3D scenes to generate novel views from the given sparse observations (Yu et al., 2021b; Chen et al., 2021; Chibane et al., 2021; Jang & Agapito, 2021; Johari et al., 2022). Regularization approaches (Seo et al., 2023a) employ a range of regularizations derived from depth supervision, patch rendering, semantic consistency, visibility, or frequency pattern (Deng et al., 2022b; Roessle et al., 2022; Guangcong et al., 2023; Niemeyer et al., 2022; Jain et al., 2021; Seo et al., 2023b; Wynn & Turmukhambetov, 2023; Somraj & Soundararajan, 2023; Yang et al., 2023; Seo et al., 2023a). Among them, (Roessle et al., 2022; Deng et al., 2022b) use the estimated depth information as supplementary supervision for more stable optimization. (Jain et al., 2021; Niemeyer et al., 2022) impose regularization on rendered patches from semantic consistency, geometry, and appearance. (Yang et al., 2023) regularizes the visible frequency range of NeRF's inputs to avoid overfitting when training starts. Other attempts include the use of cross-view pixel matching (Truong et al., 2023), cross-view feature matching (Chen et al., 2023b; Du et al., 2023), ray-entropy regularization (Kim et al., 2022), and visibility priors (Somraj & Soundararajan, 2023). Yet, no existing approach can excel across diverse complex scenes, where scene-specific heuristic adjustments are required to generate good results.

DiffusioNeRF (Wynn & Turmukhambetov, 2023) regularizes NeRF with a prior over scene geometry and color from denoising diffusion models. While also utilizing diffusion models, our approach is different from DiffusioNeRF in the following aspects: 1) DiffusioNeRF uses an unconditional generation model to generate RGBD patches, while our approach uses a conditional generation model to fine-tune artifacts whole images. 2) DiffusioNeRF leverages a trained DDM model to regularize NeRF-rendered image patches. In contrast, our method directly uses images refined by deceptive diffusion model as input to produce the fine NeRF.

**Diffusion models for view synthesis.** Recently, diffusion models (Ho et al., 2020; Nichol & Dhariwal, 2021), a powerful class of generative models that follows a Markov process to denoise inputs, have demonstrated notable success on conditional generation (Zhang & Agrawala, 2023; Rombach et al., 2022), such as text-to-image generation (Ramesh et al., 2022; Saharia et al., 2022b; Zhang & Agrawala, 2023), image super-resolution (Li et al., 2022; Saharia et al., 2022c), and inpainting (Lugmayr et al., 2022; Saharia et al., 2022a). By capitalizing on powerful 2D diffusion models, a number of works have advanced the frontier of 3D computer vision tasks, such as 3D content generation. DreamFusion (Poole et al., 2022) and Magic3D (Lin et al., 2022) perform text-guided 3D generation by optimizing a NeRF from scratch. Closer to our work, (Chen et al., 2023a; Karnewar et al., 2023; Melas-Kyriazi et al., 2023; Deng et al., 2022a; Zhou & Tulsiani, 2022; Gu et al., 2023) deal with 3D-aware conditional image generation. To achieve this, (Liu et al., 2023) uses a diffusion model trained on synthetic data as geometric priors to synthesize novel views given one single image. (Zhou & Tulsiani, 2022) transfers 3D consistent scene representation from a view-conditioned diffusion model to improve few-shot novel view synthesis. Unlike these methods that utilize diffusion models in a 3D setting, our approach does not employ them as a "scorer" for regularization. Instead, we use the images generated by the diffusion model as auxiliary pseudo-observations directly for NeRF training. As a result, our method avoids inferring the diffusion model at every training step.

## 3 METHOD

To enable plausible and 3D-consistent predictions given only sparse-view observations, we take advantage of diffusion models to "densify" the inputs using the approach illustrated in Figure 1. We first train a coarse NeRF from the input views, creating conditions for the generation of pseudo-observations (Section 3.2). Then, given the rendered RGB-D images from the coarse NeRF, we propose a deceptive diffusion model (Section 3.3) to refine these images into pseudo-observations. We use these plausible pseudo-observations to supplement the input views and train a fine NeRF using a progressive training strategy(Section 3.4).

### 3.1 BACKGROUND

**Neural Radiance Fields.** A radiance field is a continuous function $f$ mapping a 3D coordinate $\mathbf{x} \in \mathbb{R}^3$ and a viewing directional unit vector $\mathbf{d} \in \mathbb{S}^2$ to a volume density $\sigma \in [0, \infty)$ and RGB values $\mathbf{c} \in [0, 1]^3$. A neural radiance field (NeRF) (Mildenhall et al., 2020) uses a multi-layer

perceptron (MLP) to parameterize this function:

$$f_\theta : (\mathbf{x}, \mathbf{d}) \mapsto (\sigma, \mathbf{c}) \tag{1}$$

where $\theta$ denotes MLP parameters. While existing NeRF variants employ explicit voxel grids (Yu et al., 2021a; Fridovich-Keil et al., 2022; Chen et al., 2022) instead of MLPs to parameterize this mapping for improved efficiency, our proposed approach is compatible with both MLP-based NeRFs and voxel grid-based variants.

**Volume Rendering.** Rendering each image pixel given a neural radiance field $f_\theta$ involves casting a ray $\mathbf{r}(t) = \mathbf{o} + t\mathbf{d}$ from the camera center $\mathbf{o}$ through the pixel along direction $\mathbf{d}$. The predicted color for the corresponding pixel is computed as:

$$\hat{\mathbf{C}} = \sum_{k=1}^{K} \hat{T}(t_k)\alpha(\sigma(t_k)\delta_k)\mathbf{c}(t_k), \tag{2}$$

where $\hat{T}(t_k) = \exp\left(-\sum_{k'=1}^{k-1} \sigma(t_k)\delta(t_k)\right)$, $\alpha(x) = 1 - \exp(-x)$, and $\delta_p = t_{k+1} - t_k$. A vanilla NeRF is optimized over a set of input images and their camera poses by minimizing the mean squared error (photometric loss):

$$\mathcal{L}_{\text{pho}} = \sum_{\mathbf{r} \in \mathcal{R}} \|\hat{\mathbf{C}}(\mathbf{r}) - \mathbf{C}(\mathbf{r})\|_2^2 \tag{3}$$

## 3.2 COARSE NeRF FROM SPARSE INPUTS

Given only a few observations of a scene, i.e., input images $\{\mathbf{C}_{\text{input}}^i\}$ with associated viewpoints $\{\phi_{\text{input}}^i\}$, Using these sparse inputs, we first train an initial coarse NeRF, denoted by $\text{R}_{\text{coarse}}$, to obtain a rough reconstruction of the scene. The goal of this coarse NeRF reconstruction is to generate initial RGB images and depth predictions at novel views, which will be used as control images feeding into the deceptive diffusion model to generate pseudo-observations at the same viewpoints.

To avoid NeRF's over-fast convergence on high-frequency components of inputs, we use a linearly increasing frequency mask to regulate the visible frequency spectrum based on the training time steps (Yang et al., 2023). We randomly sample novel views $\{\phi_{\text{pseudo}}^i\}$ within a bounding box defined by the outermost input views and render corresponding RGB-D images with $\text{R}_{\text{coarse}}$:

$$(\hat{\mathbf{C}}_{\text{coarse}}^i, \hat{\mathbf{D}}_{\text{coarse}}^i) = \text{R}_{\text{coarse}}(\phi_{\text{pseudo}}^i). \tag{4}$$

Although the resulting synthesized images and depth maps still exhibit inevitable and obvious artifacts, they can provide some good guidance for the deceptive diffusion model to obtain refined novel view images as plausible pseudo-observations.

## 3.3 DECEPTIVE DIFFUSION MODEL

We propose a 2D diffusion model $\text{G}$ that conditions on a coarse RGB image $\hat{\mathbf{C}}_{\text{coarse}}$ and its corresponding depth prediction $\hat{\mathbf{D}}_{\text{coarse}}$ from $\text{R}_{\text{coarse}}$ to synthesize a refined natural image (pseudo-observation) $\hat{\mathbf{C}}_{\text{pseudo}}$ from the same viewpoint:

$$\hat{\mathbf{C}}_{\text{fine}} = \text{G}(\hat{\mathbf{C}}_{\text{coarse}}, \hat{\mathbf{D}}_{\text{coarse}}), \tag{5}$$

where $\text{G}$ in essential rectifies images from the coarse NeRF and is thus termed the deceptive diffusion model. The photo-realistic natural images generated serve as plausible pseudo-observations to cover scarcely observed regions.

Our approach capitalizes on latent diffusion models (Rombach et al., 2022), which leverages natural image priors derived from internet-scale data to help ameliorate unnaturalness caused by few-shot NeRFs. Artifacts generated by NeRFs often float in empty space and are therefore highly conspicuous in depth prediction. To provide additional guidance, we condition this process on NeRF's depth predictions.

To this end, given a dataset of triplets $\left\{\left(\mathbf{C}_{\text{fine}}^i, \mathbf{C}_{\text{coarse}}^i, \mathbf{D}_{\text{coarse}}^i\right)\right\}$, we fine-tune a pre-trained diffusion model, consisting of a latent diffusion architecture with an encoder $\mathcal{E}$, a denoiser U-Net $\epsilon_\theta$, and a decoder $\mathcal{D}$. We solve for the following objective to fine-tune the model:

$$\min_\theta \mathbb{E}_{z\sim\mathcal{E}, t, \epsilon\sim\mathcal{N}(0,1)} \|\epsilon - \epsilon_\theta(z_t, t, c(\mathbf{C}_{\text{coarse}}, \mathbf{D}_{\text{coarse}}, s))\|_2^2, \quad (6)$$

where the diffusion time step $t \sim [1, 1000]$ and $c(\mathbf{C}_{\text{coarse}}, \mathbf{D}_{\text{coarse}}, s)$ is the embedding of the coarse RGB image, depth estimation, and a text embedding $s$ of the coarse image.

**Text embedding.** To derive a text embedding from the input coarse NeRF image, we first generate a text prompt $s_0$ using a pre-trained image captioning network. While image captioning reliably provides descriptive textual representations for most coarse NeRF images, its efficacy can diminish for images of lower quality or those with pronounced artifacts. To counteract this, we adopt the textual inversion (Gal et al., 2022). We optimize a shared latent text embedding $s_*$ shared by all the input observations and coarse NeRF images. By concatenating the embeddings we formulate a composite feature $s = [s_0, s_*]$ that encapsulates both the semantic and visual attributes of the input image. This combined strategy not only ameliorates the shortcomings of image captioning but also ensures the stylistic congruence of the generated pseudo-observations with the input images.

**Effective control upon diffusion models.** To enable large pre-trained diffusion models (e.g., Stable Diffusion) to refine RGB-D renderings from coarse NeRFs and synthesize photo-realistic pseudo-observations, we fine-tune them conditioned on the coarse NeRF RGB-D renderings. To enable diffusion models to learn such specific input conditions without disrupting their prior for natural images, we leverage ControlNet (Zhang & Agrawala, 2023) to efficiently implement the training paradigm discussed below while preserving the production-ready weights of pre-trained 2D diffusion models.

**Data augmentation for the deceptive diffusion model.** To enable the deceptive diffusion model to generate an artifact-free image from the same viewpoint with the coarse NeRF's rendered RGB image and depth map, we need to construct a dataset of triplets $\left\{\left(\mathbf{C}_{\text{fine}}^i, \mathbf{C}_{\text{coarse}}^i, \mathbf{D}_{\text{coarse}}^i\right)\right\}$. Specifically, this is achieved by training two versions of NeRF for the same scene: a fine version of NeRF trained on all images and a coarse version of NeRF trained on only one-fifth of the images.

By rendering from the same viewpoint, such a coarse-fine NeRF duo can render paired training samples. However, due to limited computational resources, we cannot afford to conduct NeRF duos training across a plethora of scenarios. Therefore, as illustrated in Figure 2, we introduce a data augmentation paradigm to mitigate the computational cost associated with preparing training data. Rather than exclusively relying on image pairs derived from NeRF duos, we exploit a more straightforward data source during the initial phase of training. We add random Gaussian noise to RGB images, utilizing these noised images and accompanying depth maps as training inputs, while retaining the original RGB images as the training objectives. In this manner, we can readily obtain training samples by simply pairing RGB-depth data and introducing noise. Following the initial stage, we revert to employing coarse-fine image pairs

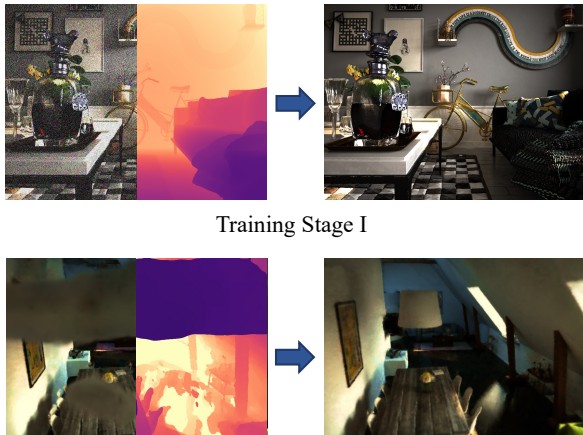

Training Stage I

Training Stage II

Figure 2: **Data augmentation for the deceptive diffusion model.** In the first stage, we augment the training samples by using noisy RGB images and depth maps as inputs, and the denoised RGB images as training targets. In the second stage, we use coarse NeRF RGB images and depth maps as inputs, and fine NeRF RGB images from the same viewpoint as training targets.

synthesized by opposing NeRFs during the subsequent phase of training. While there is a discernible distinction between the two stages, the first stage adeptly equips our deceptive diffusion model with

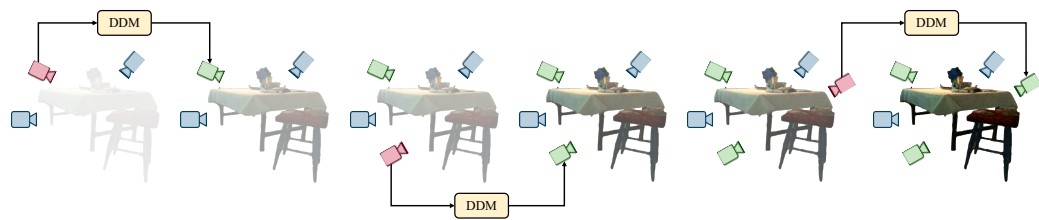

Figure 3: **Progressive Deceptive-NeRF training.** At each iteration, we use the current NeRF renderings to create pseudo-observations to enhance the next iteration's NeRF training.

the necessary prior knowledge to estimate RGB images based on depth maps (with the guidance of imperfect RGB images).

### 3.4 FINE NeRF WITH PSEUDO-OBSERVATIONS

Using the deceptive diffusion model, we obtain plausible pseudo-observations of the scene, denoted as $\{\mathbf{C}^i_{\text{pseudo}}\}$. Thanks to the natural image prior from the latent diffusion model, the pseudo-observations can eliminate the artifacts in the images rendered by the coarse NeRF. As our final 3D representation of the scene, we train a fine NeRF $\text{R}_{\text{fine}}$ model by combining the original input images (real) and pseudo-observations (fake). Given that pseudo-observations generated by the deceptive diffusion model can sometimes be inconsistent with input images, we adopt a strategy of differential selection. Specifically, we sample twice the number of required pseudo-observations for $\{\phi^i_{\text{pseudo}}\}$ and generate corresponding fine images for all of them. We then select the top 50% with the highest perceptual similarity to input images, quantified through the LPIPS metric, for fine NeRF training.

In doing so, Deceptive-NeRF alleviates the struggle of NeRF in the face of sparse observations by synthesizing fake but plausible observations. It should be noted that because the deceptive diffusion model does not constrain cross-view consistency when synthesizing images, inconsistencies may exist between the pseudo-observations and the input images. However, we found that such inconsistencies were automatically corrected during the training of the fine NeRF.

Despite general improvement in the rendering quality with the procedure discussed above, we identified that there exists a potential pitfall where the generated details might not completely align with the real scenario. To mitigate this issue, we propose a progressive training scheme as illustrated in Figure 3: In each iteration, we sample new viewpoints and use the current NeRF to render the RGB and depth maps. Then, the deceptive diffusion model generates pseudo-observations from these renderings. Enhancing existing observation sets with pseudo-observations, we train a new NeRF for the next iteration.

## 4 EXPERIMENTAL RESULTS

In this section, we evaluate our proposed Deceptive-NeRF method both qualitatively and quantitatively across a variety of challenging scenarios. We present comparisons of our model with state-of-the-art approaches and conduct an analysis of the building components of our approach. Please refer to our supplementary document and video for additional experimental results.

### 4.1 EXPERIMENTAL SETTINGS

**DDM training.** Our dataset for training the deceptive diffusion model is derived from Hypersim (Roberts et al., 2021). Hypersim contains 461 photorealistic synthetic indoor scenes and 77,400 images associated with depth maps. In the first stage, we corrupt 60,000 images by adding additive Gaussian noise with a standard deviation of 0.3. We use these noisy images and their depth maps as training input and the original images as training targets. For the second stage, we train coarse and fine NeRF duos for the same scenes, where coarse NeRFs are trained only with one-fifth of the images. Coarse NeRFs render RGB images and depth maps as training inputs while fine NeRFs render fine RGB images from the same viewpoints as training targets. We use 40 scenes and 2,000 images for the data generation of this stage. With such a dataset, we fine-tune a pre-trained Stable Diffusion model with ControlNet (Zhang & Agrawala, 2023) into our deceptive diffusion model. We set four control map channels to match the RGB-D inputs and use all default parameters for the

| DietNeRF | FreeNeRF | DiffusioNeRF | Ours | Ground Truth |
| --- | --- | --- | --- | --- |

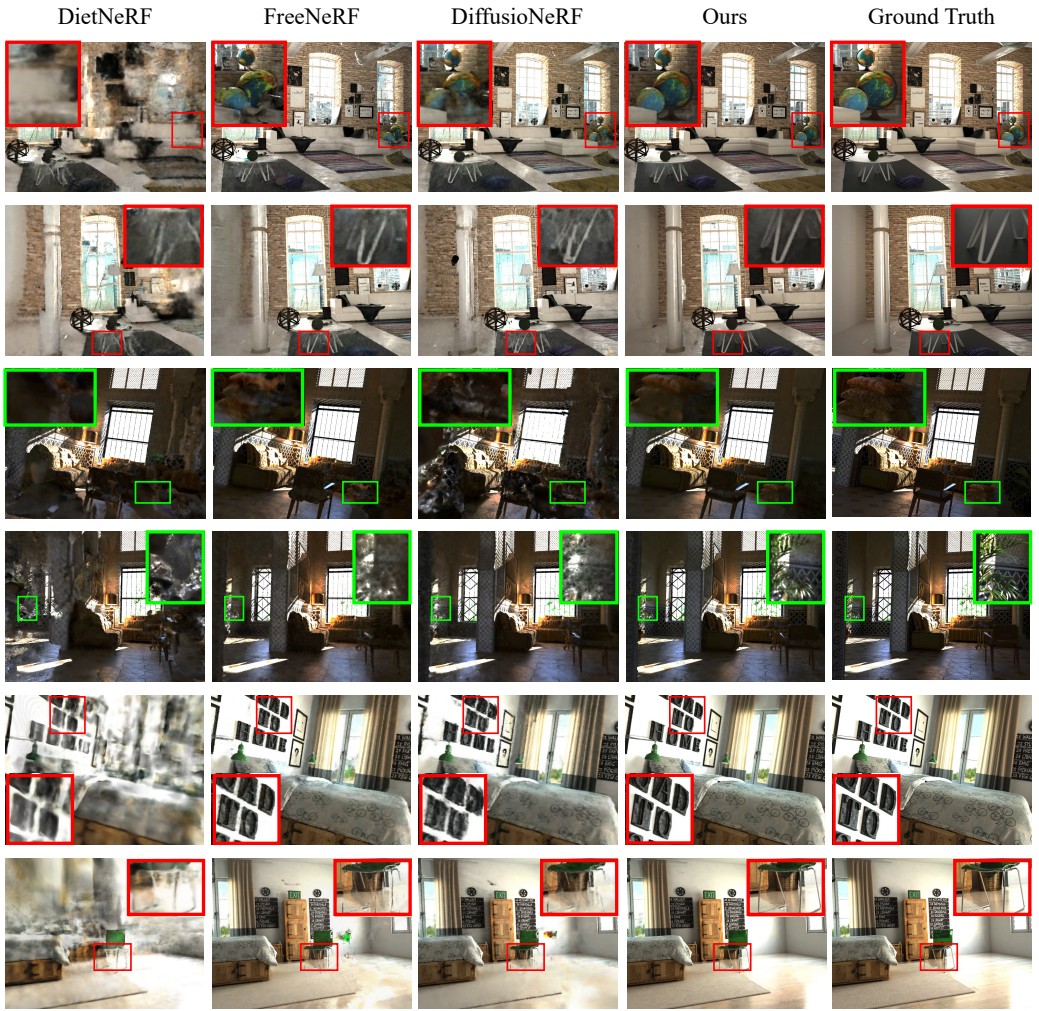

Figure 4: **Qualitative comparison on Hypersim.** Our Deceptive-NeRF synthesizes novel views with fewer artifacts, while baseline approaches tend to produce unreasonable reconstructions or floating artifacts. Zoom in for a detailed comparison.

fine-tuning task. We resize all images to a resolution of 512x512. We finetune our model on a single NVIDIA GeForce RTX 3090 Ti GPU for 10 days.

**Deceptive-NeRF implementation details.** For both the coarse and fine NeRF models, we adopt the Nerfacto method from NerfStudio (Tancik et al., 2023) as the backbone, utilizing the default proposal sampling, scene contraction, and appearance embeddings. We set $N_{\text{iter}} = 3$ for our progressive training strategy. We set the total number of synthesized pseudo-observations to be twice the number of input views, and at each iteration, we generate $\frac{\#\text{pseudo-obs.}}{\#\text{iterations}}$ of them. At each iteration, we double the number of generated observations and discard the defective 50%. We randomly sample novel views $\{\phi_{\text{pseudo}}^{i}\}$ within the bounding box defined by the outermost input cameras. Based on our experiments, we discuss the computational consumption of our approach in the appendix.

**Datasets and Metrics.** We evaluate the performance of our Deceptive-NeRF method and the baseline methods on the Hypersim (Roberts et al., 2021), as and LLFF (Mildenhall et al., 2019) datasets. Hypersim presents a challenging benchmark for few-shot indoor scene novel view synthesis. We assess different approaches using scenes that were held out from our DDM training dataset. While LLFF has been extensively adopted for evaluating novel view synthesis algorithms, the dataset features mostly forward-facing scenes and are less challenging, where Deceptive-NeRF and existing competitive approaches perform comparably well. Thus the relevant LLFF results are

Table 1: **Quantitative comparison on Hypersim.** `best` `second-best` `third-best`

| Method | PSNR(↑) | | | SSIM(↑) | | | LPIPS(↓) | | |
|---|---|---|---|---|---|---|---|---|---|
| | 5-view | 10-view | 20-view | 5-view | 10-view | 20-view | 5-view | 10-view | 20-view |
| Mip-NeRF 360 | 10.73 | 13.28 | 14.41 | 0.239 | 0.250 | 0.511 | 0.593 | 0.566 | 0.549 |
| PixelNeRF | 7.76 | 8.31 | 10.90 | 0.221 | 0.380 | 0.374 | 0.542 | 0.571 | 0.503 |
| MVSNeRF | 11.58 | 12.00 | 14.42 | 0.271 | 0.274 | 0.315 | 0.563 | 0.512 | 0.457 |
| DS-NeRF | 13.79 | 13.66 | 18.80 | 0.388 | 0.431 | 0.488 | 0.515 | 0.511 | 0.481 |
| DietNeRF | 13.01 | 13.51 | 18.62 | 0.417 | 0.479 | 0.481 | 0.541 | 0.527 | 0.472 |
| RegNeRF | 15.65 | 18.59 | 19.26 | 0.491 | 0.501 | 0.519 | 0.516 | 0.451 | 0.362 |
| DiffusioNeRF | 16.40 | 17.22 | 19.88 | 0.451 | 0.470 | 0.656 | 0.432 | 0.404 | 0.416 |
| FlipNeRF | 15.43 | 17.47 | 19.36 | 0.456 | 0.569 | 0.585 | 0.350 | 0.415 | 0.312 |
| FreeNeRF | 17.20 | 18.06 | 20.20 | 0.599 | 0.671 | 0.706 | 0.431 | 0.286 | 0.237 |
| Ours | 18.85 | 19.86 | 21.21 | 0.649 | 0.724 | 0.765 | 0.326 | 0.296 | 0.227 |

deferred to the supplementary material. We quantitatively analyze our approach and baselines using three metrics, including peak signal-to-noise ratio (PSNR), structural similarity index measure (SSIM) (Wang et al., 2004), mean absolute error (MAE), and learned perceptual image patch similarity (LPIPS) (Zhang et al., 2018). All quantitative results reported are computed by averaging held-out testing views (different from all input views as well as pseudo-observations). Furthermore, to demonstrate the generalization ability of our trained model on indoor scenes beyond Hypersim, we report experimental results on the ScanNet (Dai et al., 2017) and 3D-FRONT (Fu et al., 2021) datasets in the appendix.

**Baselines.** We compare our method with several methods within a similar scope. Among them, mip-NeRF 360 (Barron et al., 2022) stands as a state-of-the-art general NeRF model. PixelNeRF (Yu et al., 2021b), MVSNeRF (Chen et al., 2021), and SRF (Chibane et al., 2021) are representative pre-trained methods, exploiting the DTU and LLFF datasets for pre-training. We also compare our approach against diverse regularization approaches, including DS-NeRF (Deng et al., 2022b), Diet-NeRF (Jain et al., 2021), RegNeRF (Niemeyer et al., 2022), DiffusioNeRF (Wynn & Turmukhambetov, 2023), FlipNeRF (Seo et al., 2023a), and FreeNeRF (Yang et al., 2023). We also consider neural implicit surface reconstruction approaches (Yu et al., 2022a; Oechsle et al., 2021; Wang et al., 2021; Yu et al., 2022b). However, as they prioritize accurate reconstruction of object surface and require dense observations (typically around 100 input views), they fail to give reasonable results in our few-shot setting.

## 4.2 COMPARISON

In Table 1, we present the quantitative results. Our Deceptive-NeRF outperforms competing methods across almost all the evaluated metrics. Specifically, for the 5-view and 20-view settings, our approach is superior in every metric. In the 10-view setting, Deceptive-NeRF achieves the highest PSNR and SSIM and only ranks second in LPIPS. For a visual comparison, we provide qualitative results of our approach and baselines on the Hypersim dataset with 5 input views in Figure 4. While other methods can produce reasonable novel view renderings, Deceptive-NeRF excels in capturing object-level details. Our results aren't marred by the ambiguous pixels observed in the outputs of competing methods.

## 4.3 ABLATON STUDY

We conduct ablation studies on the following design choices using the Hypersim dataset under the 20-view setting: **1) Progressive Training.** To assess the effectiveness of our progressive training strategy, we experiment with a variant of our method that omits progressive training. This variant directly generates all pseudo-observations and employs them to train a fine NeRF, which then serves as the final scene representation. **2) Depth Conditioning.** Our deceptive diffusion model generates pseudo-observations conditioned on rendered depth maps. To gauge the significance of this choice, we train a variant that solely conditions on raw RGB images for generating pseudo-observations. **3) Data Augmentation.** We evaluate the impact of our data augmentation procedure when training our deceptive diffusion model. Specifically, we train the model without the initial stage and rely solely on coarse-fine NeRF pairs to generate training samples. **4) Text Embedding.** Our approach to

Table 2: **Quantitative ablation study.** best    second-best    third-best    supplement

| Progressive | Depth | Two-stage | Caption | Inversion | Filtering | PSNR (↑) | SSIM (↑) | LPIPS (↓) |
|---|---|---|---|---|---|---|---|---|
|  | ✓ | ✓ | ✓ | ✓ | $L$ | 19.90 | 0.555 | 0.358 |
| ✓ |  | ✓ | ✓ | ✓ | $L$ | 18.79 | 0.489 | 0.352 |
| ✓ | ✓ |  | ✓ | ✓ | $L$ | 20.49 | 0.619 | 0.290 |
| ✓ | ✓ | ✓ |  | ✓ | $L$ | 21.59 | 0.758 | 0.236 |
| ✓ | ✓ | ✓ | ✓ |  | $L$ | 20.58 | 0.744 | 0.239 |
| ✓ | ✓ | ✓ | ✓ | ✓ | $L$ | 22.41 | 0.812 | 0.202 |
| ✓ | ✓ | ✓ | ✓ | ✓ | $W$ | 22.37 | 0.811 | 0.217 |

Ground Truth    w/o progressive    w/o depth    w/o two-stage    w/o textual inversion    Ours

Figure 5: **Qualitative ablation study.** Our full model synthesizes novel views with fewer artifacts and finer details.

text embedding integrates both image captioning and textual inversion. This combination addresses severely artifacted images while ensuring stylistic consistency. We test two variants of our model, one without image captioning and the other without textual inversion. As illustrated in Figure 5 and Table 2, our complete model synthesizes the most photorealistic novel views and outperforms other methods in all quantitative metrics. **5) Filtering.** We employ a top-50% filtering strategy to discard inconsistent pseudo-observations. In addition to using the LPIPS metric ("$L$") to measure the perceptual similarity between input images and pseudo-observations, we also experiment with the confidence score of NeRF-W ("$W$") (Martin-Brualla et al., 2021; Talebi & Milanfar, 2018). The final reconstruction results obtained through these two strategies are close.

In addition, we provide quantitative and qualitative evaluations of pseudo-observations in the appendix.

# 5 DISCUSSION

**Limitations.** While leveraging 2D diffusion models to enhance 3D neural representations in a novel manner, our approach faces several limitations. First, the pseudo-observations generated by the deceptive diffusion model are not guaranteed to accurately reflect ground truth. Consequently, our results may appear deceptively realistic yet incorrect. Furthermore, Deceptive-NeRF is still dealing with a reconstruction problem and not capable of generating 3D content from scratch. In our experiments, since we trained the model on the Hypersim dataset (indoor scenes), performance degradation was observed when generalizing it to general real-world scenes. In the future, our approach holds promise for generalization to a broader range of scenes, particularly as large-scale datasets encompassing general 3D scenes become increasingly available in the future.

**Conclusion.** We introduce Deceptive-NeRF, which synthesizes plausible pseudo-observations for improving NeRF reconstruction from sparse input. A coarse NeRF model is first reconstructed from the given sparse input and subsequently renders coarse novel views. Our deceptive diffusion models turn novel views rendered by the coarse NeRF into pseudo-observations. The deceptive diffusion model generates pseudo-observations that faithfully preserve the semantics underlying the given scene while consistent with the sparse inputs. Finally, we use pseudo-observations to produce a high-quality reconstruction with a progressive NeRF training strategy. Extensive experiments and comparisons demonstrate that our method is effective and can generate perceptually high-quality NeRF reconstructions even with highly sparse inputs.

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

APPENDIX

## A  VIDEO

For better visualization of our reconstruction results, we create a set of video visualizations with free-viewpoint rendering. We highly recommend to watch *supplementary_video.mp4* for more results.

## B  PSEUDO-CODE FOR PROGRESSIVE DECEPTIVE-NERF TRAINING

---
**Algorithm 1** Progressive Deceptive-NeRF Training
---
1: **Input:** Images $\mathbf{C}_{\text{input}}$ with associated camera poses $\phi_{\text{input}}$
2: $\mathbf{C} \leftarrow \mathbf{C}_{\text{input}}$
3: $\phi \leftarrow \phi_{\text{input}}$
4: $\text{NeRF}_{\text{current}} \leftarrow \text{TRAINNERF}(\mathbf{C}, \phi)$
5: **for** $i = 1$ to $N_{\text{iter}}$ **do**
6:     $\phi_{\text{pseudo}} \leftarrow \text{SAMPLENOVELVIEW}(\phi)$
7:     $\mathbf{C}_{\text{coarse}}, \mathbf{D}_{\text{coarse}} \leftarrow \text{RENDERNERF}(\text{NeRF}_{\text{current}}, \phi_{\text{pseudo}})$
8:     $\mathbf{C}_{\text{fine}} \leftarrow \text{RECTIFY}(\mathbf{C}_{\text{coarse}}, \mathbf{D}_{\text{coarse}})$
9:     $\mathbf{C}_{\text{fine}} \leftarrow \text{DISCARDDEFECTIVE}(\mathbf{C}_{\text{fine}})$
10:    $\text{APPEND}(\mathbf{C}, \mathbf{C}_{\text{fine}})$
11:    $\text{APPEND}(\phi, \phi_{\text{pseudo}})$
12:    $\text{NeRF}_{\text{current}} \leftarrow \text{TRAINNERF}(\mathbf{C}, \phi)$
13: $\text{NeRF}_{\text{final}} \leftarrow \text{NeRF}_{\text{current}}$
---

## C  COMPUTATIONAL CONSUMPTION

**Fine-Tuning Diffusion Model.** Our model was fine-tuned using a single NVIDIA GeForce RTX 3090 Ti GPU over a period of 10 days. Please kindly note that this timeframe was achieved with personal computer-level resources. Enhanced computational resources and advanced techniques Hu et al. (2022) should significantly reduce this duration. Furthermore, this is a one-time process, allowing the adapted model to be applied directly to per-scene NeRF training without further fine-tuning.

**Per-Scene NeRF Training.** With the fine-tuned diffusion model, we conducted per-scene NeRF training on a single NVIDIA GeForce RTX 3090 Ti GPU. The purpose of the process is twofold: 1) training the radiance field and 2) generating pseudo-observations with the diffusion model. For example, in an experiment on Hypersim with 10 training views, our progressive training approach produced 40 images (50% discarded due to defects), taking 3 minutes. The bulk of the time (42 minutes) was for radiance field training, with progressive training at $N_{\text{iter}} = 3$ resulting in $4 \times 30000$ training steps. The time spent on both aspects is summarized in Table A:

Table A: **Time analysis for per-scene NeRF training.**

|  | Synthesizing Pseudo-Observations | Radiance Fields Training |
|---|---|---|
| Time | 3 minutes | 42 minutes |

A comparative analysis of the time with and without progressive strategy is presented below. The times reported encompass both pseudo-observation generation and serial radiance field optimization.

Table B: **Comparison of Runtime with Progressive and without Progressive Training.**

|  | w/ progressive | w/o progressive |
|---|---|---|
| Time | 45 minutes | 22 minutes |

Despite integrating a diffusion model and serial radiance field training, our approach does not markedly increase runtime. The diffusion model's role is limited to a mere 3 minutes for pseudo-observation synthesis. This is in contrast to recent studies that combine NeRF and diffusion models, which necessitate diffusion model inference at each NeRF training step. This efficiency significantly reduces computational overhead. In fact, our method is faster than most baselines, which generally exceed an hour. (Since various methods implement upon different frameworks in their official codes, we refrain from a formal runtime comparison, providing this only as a reference.)

# D    EVALUATION OF PSEUDO-OBSERVATIONS

In this section, we qualitatively and quantitatively assess our generated pseudo-observations and validate the effectiveness of our deceptive diffusion model. For comparison, given coarse NeRF-generated images, we use image restoration models instead of our deceptive diffusion model to synthesize pseudo-observations. The experimental setup follows that of Section 4.3. The image restoration model we adopt is Restormer (Zamir et al., 2022), a popular image restoration model with state-of-the-art performance. Its officially released code provides models for a number of image restoration tasks including image denoising, image deraining, motion deblurring, and defocus deblurring.

In the qualitative results shown in Figure A, pseudo-observations synthesized by our deceptive diffusion model demonstrate effective removal of the floating artifacts caused by sparse observation in coarse NeRFs, also mitigating blurriness. In contrast, image restoration models fail to convert coarse NeRF images into reasonable pseudo-observations. This is primarily because they are designed and trained for specific image restoration tasks, not for generating pseudo-observations.

In Table C, we report quantitative results, including the similarity between generated pseudo-observations and their ground truth (measured in terms of p-PSNR, p-SSIM, and p-LPIPS), as well as the performance of the complete model on the novel view synthesis task. The model employing our deceptive diffusion model synthesizes the most photo-realistic pseudo-observations and achieves the best final performance in novel view synthesis.

In summary, our deceptive diffusion model, specifically tailored for generating pseudo-observations from coarse NeRF images and utilizing depth and texture cues, effectively eliminates floating artifacts and blurriness in coarse NeRF images, thereby producing natural pseudo-observations.

Table C: **Quantitative evaluation of pseudo-observations.** The model with our deceptive diffusion model synthesizes the most photo-realistic pseudo-observations and achieves the best performance in novel view synthesis.

| Method | p-PSNR (↑) | p-SSIM (↑) | p-LPIPS (↓) | PSNR (↑) | SSIM (↑) | LPIPS (↓) |
|---|---|---|---|---|---|---|
| Denoising | 16.39 | 0.716 | 0.325 | 18.77 | 0.716 | 0.274 |
| Deraining | 16.11 | 0.707 | 0.338 | 18.48 | 0.712 | 0.271 |
| Motion Deblurring | 16.04 | 0.705 | 0.324 | 19.01 | 0.701 | 0.251 |
| Defocus Deblurring | 16.13 | 0.691 | 0.334 | 18.20 | 0.724 | 0.285 |
| Deceptive Diffusion Model (Ours) | **19.24** | **0.754** | **0.256** | **19.85** | **0.770** | **0.231** |

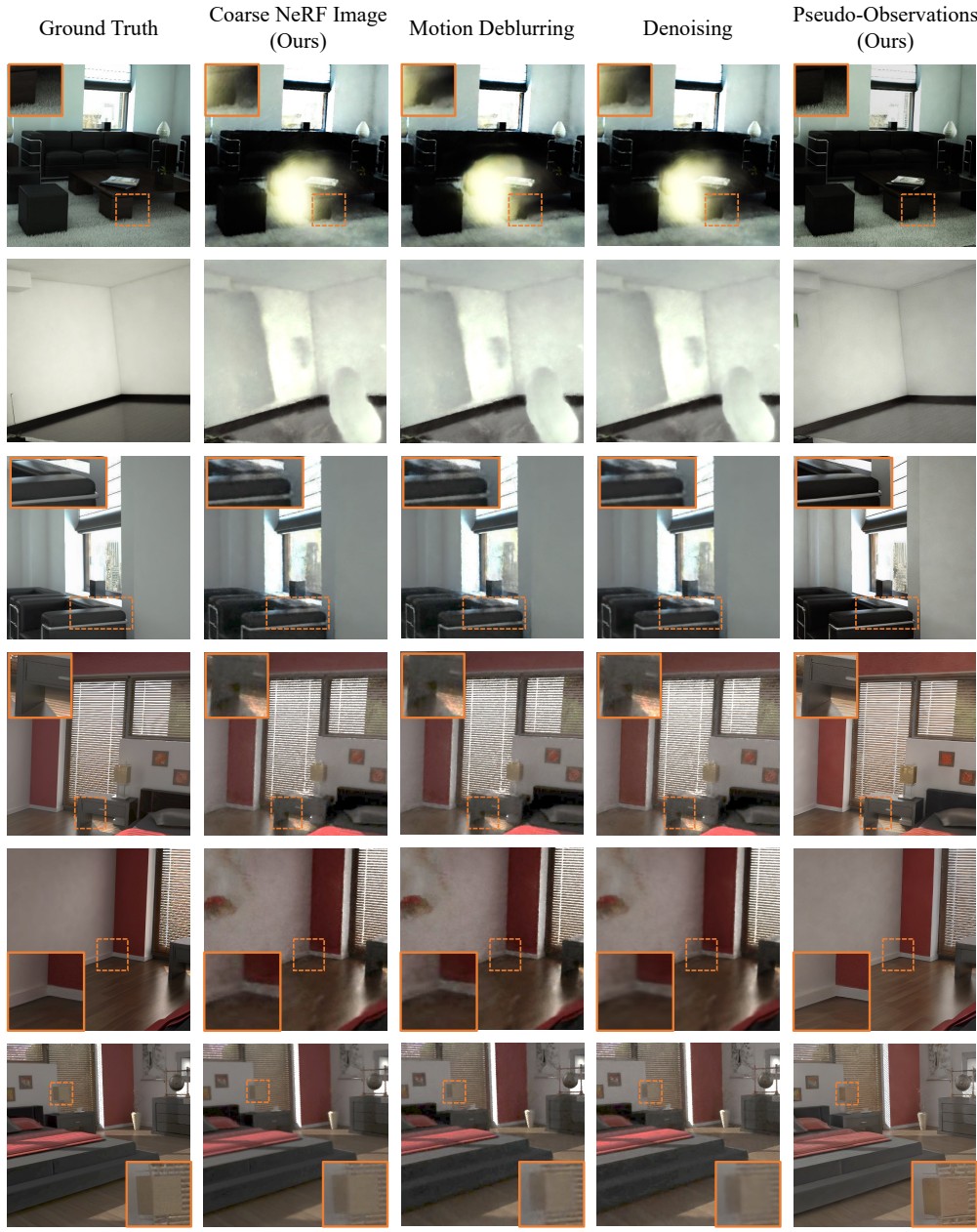

| Ground Truth | Coarse NeRF Image (Ours) | Motion Deblurring | Denoising | Pseudo-Observations (Ours) |

Figure A: **Qualitative evaluation of pseudo-observations.** Pseudo-observations synthesized by our deceptive diffusion model can remove the floating artifacts and blurriness caused by sparse observation in coarse NeRFs.

# E EXPERIMENTAL RESULTS ON 3D-FRONT AND SCANNET

To verify the generalization ability of our method, we evaluate our model trained on Hyper-Sim (Roberts et al., 2021) on synthetic (3D-FRONT (Fu et al., 2021)) and real (ScanNet (Dai et al., 2017)) indoor datasets using 10 input views and compared it with FreeNeRF (Yang et al., 2023).In the qualitative results presented in Figure B, our approach better recovers objects in

the scene, such as the toy and the sink and produces fewer artifacts than the baseline approach. In Table D, our method also performs better in terms of all metrics.

Our experimental results demonstrate the generalization ability of our method. The deceptive diffusion model trained on indoor data can perform well on other indoor datasets. Our method has the potential to generalize to more general 3D scenes beyond indoor settings if large-scale 3D data becomes available in the future.

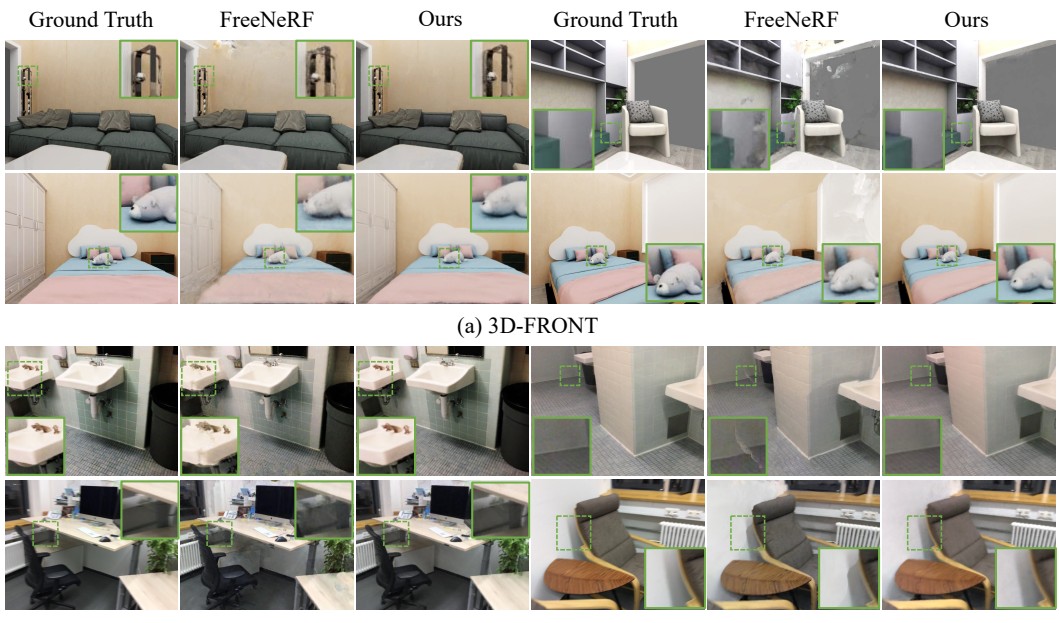

Figure B: **Qualitative comparison on 3D-FRONT and ScanNet.** Our approach better recovers objects in the scene, such as the toy and the sink with fewer artifacts.

Table D: **Quantitative comparison on 3D-FRONT and ScanNet.** Our method performs better in terms of all metrics.

| Method | 3D-FRONT | | | ScanNet | | |
|---|---|---|---|---|---|---|
| | PSNR (↑) | SSIM (↑) | LPIPS (↓) | PSNR (↑) | SSIM (↑) | LPIPS (↓) |
| FreeNeRF | 23.09 | 0.807 | 0.237 | 19.41 | 0.677 | 0.306 |
| Ours | **25.09** | **0.827** | **0.168** | **22.55** | **0.745** | **0.198** |

## F  EXPERIMENTAL RESULTS ON LLFF

We display qualitative comparisons on LLFF in Figure C and quantitative comparisons in Table E. Since the model was trained on the Hypersim dataset (indoor scenes) and was not fine-tuned on LLFF, an expected performance degradation was observed. However, our method still achieves results that are either outperforming or on par with state-of-the-art approaches. Looking ahead, our approach has the potential to generalize to more diverse scenes as large-scale datasets of general 3D scenes become more accessible in the future.

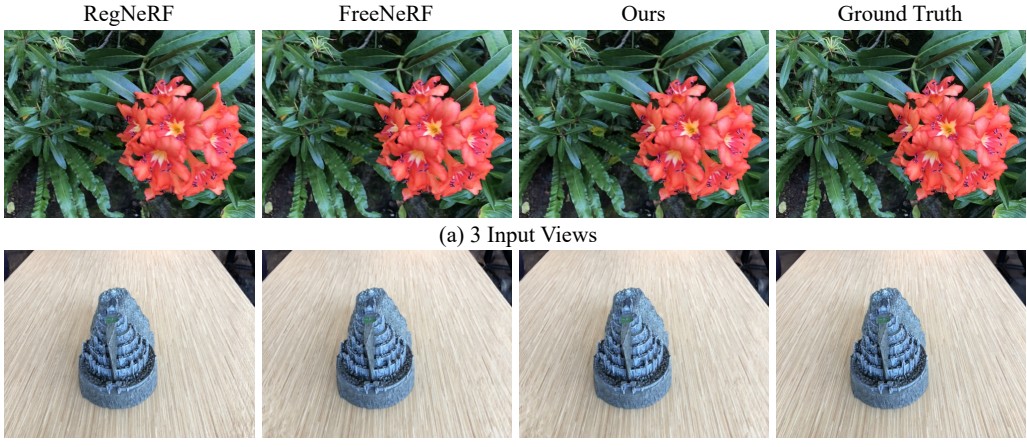

| RegNeRF | FreeNeRF | Ours | Ground Truth |

(a) 3 Input Views

(b) 6 Input Views

Figure C: **Qualitative comparison on LLFF.**

Table E: **Quantitative comparison on LLFF.** best  second-best  third-best

| Method | PSNR(↑) | | | SSIM(↑) | | | LPIPS(↓) | | |
|---|---|---|---|---|---|---|---|---|---|
| | 3-view | 6-view | 9-view | 3-view | 6-view | 9-view | 3-view | 6-view | 9-view |
| SRF | 17.07 | 16.75 | 17.39 | 0.436 | 0.438 | 0.465 | 0.529 | 0.521 | 0.503 |
| PixelNeRF | 16.17 | 17.03 | 18.92 | 0.438 | 0.473 | 0.535 | 0.512 | 0.477 | 0.430 |
| MVSNeRF | 17.88 | 19.99 | 20.47 | 0.584 | 0.660 | 0.695 | 0.327 | 0.264 | 0.244 |
| mip-NeRF | 16.11 | 22.91 | 24.88 | 0.401 | 0.756 | 0.826 | 0.460 | 0.213 | 0.160 |
| DietNeRF | 14.94 | 21.75 | 24.28 | 0.370 | 0.717 | 0.801 | 0.496 | 0.248 | 0.183 |
| RegNeRF | 18.84 | 23.22 | 24.88 | 0.573 | 0.770 | 0.826 | 0.345 | 0.203 | 0.159 |
| FreeNeRF | 19.63 | 23.73 | 25.13 | 0.612 | 0.779 | 0.827 | 0.308 | 0.195 | 0.160 |
| Ours | 19.69 | 23.81 | 25.01 | 0.609 | 0.768 | 0.821 | 0.315 | 0.193 | 0.161 |

