# OpenReview forum: "Deceptive-NeRF: Enhancing NeRF Reconstruction using Pseudo-Observations from Diffusion Models"
_ICLR.cc/2024/Conference — Submitted to ICLR 2024_

### Official Review · Reviewer_zKFv · 2023-10-22

**Soundness:** 2 fair
**Presentation:** 3 good
**Contribution:** 2 fair
**Rating:** 5
**Confidence:** 4

**Summary:**

This work introduces a new method, called Deceptive-NeRF,  which leverages diffusion models to synthesize pseudo observations to improve the few-shot NeRF reconstruction. This approach first reconstructs a coarse NeRF from sparse input data, and then utilizes the coarse NeRF to render images and subsequently generates pseudo-observations based on them. Last, a refined NeRF model is trained utilizing input images augmented with pseudo-observations. A deceptive diffusion model is proposed to adeptly convert RGB images and depth maps from coarse NeRFs into photo-realistic pseudo-observations, while preserving scene semantics for reconstruction. Experiments on the synthetic Hypersim dataset demonstrate that the proposed approach is capable of synthesizing photo-realistic novel views with very sparse inputs.

**Strengths:**

This paper introduces an approach for the few-shot novel view synthesis that leverages diffusion models to generate pseudo-observations to provide training signals.

To generate photo-realistic pseudo-observations that faithfully preserve scene semantics and input view consistency, an RGB-D conditioned diffusion model is trained on a synthetic indoor scene dataset (Hypersim).

An ablation study is conducted to verify the design of the method, including progressive training, depth condition, image captioning, and textual inversion.

Results on the Hypersim dataset show that the proposed method outperforms the existing method on the few-shot setting.

**Weaknesses:**

The idea of introducing pseudo observations to enhance the reconstruction of few-shot Neural Radiance Fields (NeRF) is a promising concept. If the diffusion model can effectively generate pseudo observations that align with the data distribution of a given scene, it has the potential to improve the quality of the refined NeRF reconstructions.

One of the primary concerns is the generalization capacity of the proposed deceptive diffusion model. The real-world scenes' data distribution is often highly intricate and diverse. However, the diffusion model is only trained on a limited dataset consisting of 40 scenes and 2000 synthetic images from the Hypersim dataset during the second stage. As the primary experiment relies on the Hypersim dataset, which shares similarities with the training data, the method's performance on the real LLFF dataset is disappointing. In specific metrics and view-number configurations, it even falls short of the freeNeRF (note that the proposed method also uses the same frequency regularization as in freeNeRF). These outcomes indicate that the proposed approach struggles to generalize to the complexities of real-world scenes.

It would be valuable to include a comparative analysis between the generated pseudo observations and the ground-truth images. This could provide insights into the fidelity of the pseudo observations and their accuracy in replicating the real data.

Information regarding the optimization time required for scene reconstruction is crucial for understanding the method's practicality and efficiency. Including this information in the paper would be helpful for readers seeking to assess the computational demands of implementing this approach.

In Table 1, a more robust baseline for scene reconstruction might be considered, such as the monoSDF method that utilizes monocular depth and normal maps as additional sources of supervision. Comparing the proposed method's performance to such a strong baseline would provide a clearer picture of its relative merits and limitations.

**Questions:**

- Further discussion on the generalization of the proposed methods.
- It is important to assess the quality of the generated pseudo observations. Detailed evaluations, including visual comparisons with ground-truth data, can help validate the effectiveness of this component in improving the NeRF reconstruction.
- Discussion for the runtime?
- Stronger baseline for scene reconstruction.
 Please refer to weakness for details.

---

> ### Author Response · Authors · 2023-11-13
>
> We would like to thank the reviewer for regarding our idea as a "promising concept" and providing constructive criticism that helped us identify areas that need improvement. Below is our response to your concerns.
>
> **Generalization Capacity (Question 1).** Please refer to the discussion in the general response. We believe that our method has the potential to achieve good few-shot novel view synthesis results on any 3D scene when large-scale general 3D scene datasets are available for training. We will soon report the results of generalizing our model to other indoor datasets beyond Hypersim to demonstrate its generalization ability.
>
> **Quality of Pseudo-Observations. (Question 2)** Thank you for your thoughtful suggestion to include a comparative analysis between our generated pseudo-observations and the ground-truth images. We have been working on it and will provide these visual comparisons soon to further strengthen our arguments.
>
> **Runtime analysis (Question 3).** Please refer to the discussion in the general response and we will soon provide runtime analysis and comparison.
>
> **Stronger Baselines (Question 4).**  Thanks for pointing out our missing baselines and we'll report their results soon.

---

> ### Author Response · Authors · 2023-11-23
>
> We would like to express our gratitude once again for your precise feedback in improving our work. Please review our revised manuscript. We have further addressed your concerns as follows:
>
> **Generalization Capacity.** In Section E, we report the performance of our model, trained on the HyperSim dataset, on two additional datasets. This demonstrates that our method not only surpasses the baseline but also confirms that models trained on the HyperSim dataset are effective on other indoor datasets. As you rightly pointed out, we have emphasized in our limitations section the performance decline when models trained on HyperSim are generalized to arbitrary 3D scenes beyond indoor environments.
>
> **Runtime Analysis.** We have included a discussion about the runtime of our method, with a time breakdown. Please refer to Section C or the general response provided above for more information.
>
>
> **Quality of Pseudo-Observations.**  We have included a comparative analysis between our generated pseudo-observations and the ground-truth images in Section D of our revised manuscript. Additionally, as suggested by Reviewer QhhZ, we provide a comparison with image restoration models, along with quantitative results. We hope they will aid in understanding the behavior of the deceptive diffusion model.
>
> **MonoSDF.** Thank you for suggesting MonoSDF [1] as an additional baseline. We experimented with MonoSDF integrated into SDFStudio [2], but in our few-shot setting (5-20 input views), MonoSDF failed to provide reasonable reconstruction. This is likely due to the fact that MonoSDF, as a neural implicit surface reconstruction approach, prioritizes accurate reconstruction of the object surface and therefore requires dense observations to achieve reasonable results.
> The comparison in Table 1 covers most of the state-of-the-art methods in the field of few-shot novel view synthesis at the time of submission. While approaches for the task of neural implicit surface reconstruction [1,3,4] are highly relevant to our task, they have a different focus. Specifically, these methods prioritize the reconstruction of object surfaces, while our focus is on photo-realistic novel view rendering directly without recovering the surface. Additionally, neural implicit surface reconstruction often requires additional inputs such as normals, depths, or masks, in addition to RGB inputs, to ensure the accuracy of the surface reconstruction, while our method does not.
> We have revised the paper to include a discussion on MonoSDF in Section 4.1.
>
>
> **References**
> [1] Yu et al., “MonoSDF Exploring Monocular Geometric Cues for Neural Implicit Surface Reconstruction”. (NeurIPS 2022)
> [2] Yu et al., “SDFStudio: A Unified Framework for Surface Reconstruction”. (https://github.com/autonomousvision/sdfstudio)
> [3] Wang et al., “NeuS: Learning Neural Implicit Surfaces by Volume Rendering for Multi-view Reconstruction”. (NeurIPS 2021)
> [4] Oechsle et al., “UNISURF: Unifying Neural Implicit Surfaces and Radiance Fields for Multi-View Reconstruction”. (ICCV 2021)

---

### Official Review · Reviewer_beZg · 2023-10-30

**Soundness:** 2 fair
**Presentation:** 3 good
**Contribution:** 2 fair
**Rating:** 6
**Confidence:** 2

**Summary:**

This method proposed a few-shot NeRF training with pseudo samples from the diffusion models as the training corpus, and a series of training strategy that can boost the performance.

**Strengths:**

The overall idea is reasonable, effective and well elaborated.

**Weaknesses:**

1. One biggest concern of this idea is that some diffusion-synthesized samples are view-inconsistent. This method proposed a 50% filtering strategy to alleviate this issue, but I don't know whether this issue can be fully bypassed. Introducing confidence score as in NeRF-W may help.
2. Computational cost. Training a diffusion model for 10 days to further finetune a NeRF sounds inefficient to me. Also, can this finetuned CLDM be applied to any in-the-wild NeRF reconstruction dataset?

**Questions:**

Beside the concerns stated above, I wonder whether you can apply your method to other popular nerf benchmars, such as DTU and LLFF? Also, what is the key limitation of your method and under which setting will this pipeline fails.

---

> ### Author Response · Authors · 2023-11-13
>
> Thank you for your constructive feedback and for acknowledging our work as 'reasonable, effective, and well-elaborated'. Below is our response to your concerns:
>
> **Filtering Strategy (Weakness 1).** Based on our experience, our filtering strategy, together with our other designs, has to some extent alleviated the occurrence of view inconsistency. We appreciate the reviewer's suggestion to adopt the confidence score as in NeRF-W and will implement it as soon as possible and report the results.
>
> **Computational Cost (Weakness 2).** Please refer to our discussion in the general response. Please kindly note that the 10-day fine-tuning time was achieved using personal computer-level computing resources (a single NVIDIA GeForce RTX 3090 Ti GPU) and can be greatly improved.
>
> **Other benchmarks (Weakness 2 and Question).** Please refer to our discussion in the general response. We have included the experimental results on LLFF in the appendix, and our method outperforms most of the baselines. We will report our method's results on other indoor scene datasets to demonstrate its generalization ability.

---

> ### Author Response · Authors · 2023-11-23
>
> We would like to express our gratitude once again for your constructive feedback on improving our paper. Please review our updated manuscript, which includes the following revisions based on your suggestions:
>
> **Confidence Score as in NeRF-W.** We have updated Table 2 to report the performance of a filtering strategy using the confidence score used in NeRF-W [1]. This approach yielded results comparable to our original design. Specifically, following [1]’s pre-filtering step, we discarded low-quality pseudo-observations that scored below the 50% threshold in the NIMA [2] assessment.
>
> **Computational Cost.** We have discussed the computational cost of our method. Please see Section C or the general response above.
>
> **Other Benchmarks.** In response to your inquiries about “Can this finetuned CLDM be applied to any in-the-wild NeRF reconstruction dataset” and “Whether you can apply your method to other popular NeRF benchmarks”, we have evaluated our method on two additional datasets. As outlined in Section E, these evaluations demonstrate that our method not only surpasses the baseline but also confirms that models trained on the HyperSim dataset can also work effectively on other indoor datasets. In Section F, we report our method's performance on the popular NeRF benchmark LLFF. While our approach exceeds most baselines, it does exhibit a performance decline due to the domain gap from our indoor training data. Nevertheless, our method shows promise for broader generalization to diverse in-the-wild scenes, a potential that should be further realized as large-scale datasets of general 3D scenes become increasingly available.
>
>
> **References**
> [1] Martin-Brualla et al., “Neural Radiance Fields for Unconstrained Photo Collections”. (CVPR 2021)
> [2] Milanfar et al., “NIMA: Neural Image Assessment”. (IEEE TIP, 2018)

---

### Official Review · Reviewer_QhhZ · 2023-11-06

**Soundness:** 3 good
**Presentation:** 2 fair
**Contribution:** 2 fair
**Rating:** 6
**Confidence:** 3

**Summary:**

The present paper proposes a diffusion model training scheme for neural radiance field-based high-quality novel view synthesis from a small number of input views. It's core idea is to train an initial radiance field based on the small number of views, and then generate novel views in this low quality field which are subsequently enhanced using a diffusion model. The resulting higher quality views can then be used to train a higher quality neural radiance field. In terms of technical novelty, most of the substance focuses on training the enhancement diffusion model. Ideally, such training would require the availability of neural fields (both degraded and high quality) for a big number of scenes which is computationally prohibitive. Thus, the authors train the diffusion model to restore a noise corrupted input image (+ depth). The work is shown to outperform other baselines using neural fields.

**Strengths:**

* This work considers an important problem following the theme of leveraging 2d generative models for 3d.
* It outperforms its considered baselines for few-shot novel view synthesis

**Weaknesses:**

* Some of the writing requires improvements, for example in related work the first paragraph ends with the statement that NeRFs require numerous images which is followed by an entire paragraph falsifying that very statement presenting recent advances on few-sample learning with NeRFs. There are also some language issues.
* The evaluation seems insufficient. Given that the diffusion model has been trained on a denoising task to reduce computational burden, it would seem meaningful to evaluate this concept on some existing image restoration networks in comparison to the proposed fine-tuning of a diffusion model.
* the approach seems to be computationally burdensome as its iterative variant requires a sequence of radiance field learning processes.

**Questions:**

* what is meant by "we use a linearly increasing frequency mask"?
* how is the depth map obtained from the nerf?
* It is surprising to me that the optimization scheme does not result in inconsistency issues. Could the authors provide some intuition for why there is no issue?
* What is meant by "We optimize a shared latent text embedding s [...]"?

---

> ### Author Response · Authors · 2023-11-13
>
> We appreciate your thoughtful comments and the attention to detail you gave to our paper.  Below is our response to your concerns.
>
> **Writing issue (Weakness 1).** Thanks for pointing that out. We will improve writing to ensure logical flow and minimize grammatical and vocabulary errors.
>
> **Image Restoration Networks (Weakness 2).** Comparing our model with the fine-tuning process with existing image restoration networks is a reasonable approach to demonstrate the effectiveness of our proposed method. We will conduct the corresponding experiments and report the results as soon as possible.
>
> **Computational Burden (Weakness 3).** Please refer to the discussion in the general response. I will soon provide more experimental analysis on the runtime.
>
> **Frequency Mask (Question 1).** When we refer to "we use a linearly increasing frequency mask," we are describing the utilization of a technique that resembles the frequency regularization in [1]. To be specific we initiate training without any positional encoding and progressively enhance the visibility of certain frequency components as the training process advances. This strategy is employed to prevent overfitting to high-frequency elements within the input data. We will clarify this in the modified version to assist readers in understanding.
>
> **Depth from NeRFs (Question 2).** We compute the depth estimation of a pixel as
> $
>    \hat{z}(\mathbf{r}) = \sum_{k=1}^{K} w_k t_k,
> $
>    where $t_k$ are the sampling locations, and the rendering weights
> $
>    w_k = T_k (1 - \exp(-\sigma_k \delta_k)).
> $
>  $\sigma_k$ is the volume density value inferred from the NeRF MLP. $\delta_k = t_{k+1} - t_k$ is the distance between adjacent sampled points, and
> $
>    T_k = \exp\left(-\sum_{i=1}^{k-1} \sigma_i \delta_i\right).
> $
>
> **View consistency (Question 3).** In our pipeline, the diffusion model does not generate a new view image from scratch, which can easily lead to inconsistent views (as seen in many 3D diffusion generation efforts). Instead, it takes as input an image rendered from a coarse NeRF and refines it to remove artifacts. The coarse image processed by the diffusion model is consistent with other views in nature and the view consistency can still be maintained after diffusion’s refinement. Moreover, our progressive strategy involves gradually moving away from the input view to synthesize a new view. As a result, the artifacts encountered by the diffusion model at any given time are limited, reducing the difficulty of the refinement task it faces.
>
> **Shared text embedding (Question 4).** As described in [2], we optimize for a new embedding vector within the embedding space of a frozen text-to-image model. This vector represents elements that appear in each input image but cannot be directly described using natural language, such as the stylistic information of a room. This embedding can be treated like any other word and is utilized to compose novel textual queries for our generative models.
>
>
> **References**
> [1] Yang et al., “FreeNeRF: Improving Few-shot Neural Rendering with Free Frequency Regularization”.
> [2] ​​Gal et al., “An Image is Worth One Word: Personalizing Text-to-Image Generation using Textual Inversion”.

---

> > ### Comment · Reviewer_QhhZ · 2023-11-17
> >
> > Thank you for this response. I will wait for the new updates and then decide if and how much I can increase my scores. Also, it would be cool, if you could use latexdiff or at least some color emphasis in the updated version of the paper to visualize which parts changed in response to the review.

---

> > > ### Author Response · Authors · 2023-11-23
> > >
> > > We would like to thank you once again for your valuable suggestions to help revise our paper. We have updated our manuscript accordingly, highlighting the changes in blue as you suggested. We hope that the following updates adequately address your concerns:
> > >
> > >
> > > **Image Restoration Networks.** Based on your suggestions, we compared the pseudo-observations synthesized by our deceptive diffusion model with those obtained using image restoration networks [1], as detailed in Section E. Our findings indicate that image restoration models typically struggle to convert coarse NeRF images into reasonable pseudo-observations. This issue arises primarily because they are designed and trained for specific image restoration tasks. In contrast, our deceptive diffusion model, which is specifically tailored for generating pseudo-observations, effectively leverages depth and texture cues to produce more photo-realistic results.
> > >
> > >
> > >
> > >
> > > **Writing Issue.** We have revised the last sentence of the first paragraph in the related works section for a better transition into the subsequent content. Additionally, we corrected other writing issues and inaccuracies throughout the manuscript.
> > >
> > >
> > > **Computational Burden.** We have discussed the computational burden of our method in Section C and in the general response above.
> > >
> > >
> > > **References**
> > > [1] Zamir et al., “Restormer: Efficient Transformer for High-Resolution Image Restoration”. (CVPR 2022)

---

### Author Response · Authors · 2023-11-13
**General Response**

We thank all the reviewers for their time and constructive reviews. We are encouraged that the reviewers appreciate our attempt toward leveraging 2D generative models for improving few-shot NeRF. We first address common concerns, followed by detailed responses to individual reviewers. In order to have a more efficient discussion, we will first answer questions that do not require additional experiments and will report new experimental results one after another during the discussion.

We would appreciate it very much that reviewers would consider raising the scores if the following, including suggested experiments done before the end of the discussion period and fixing of exposition issues, can adequately address your concerns.

**Time Analysis and Computational Cost (Weakness 3 from Reviewer QhhZ, Weakness 2 from Reviewer beZg, and Question 3 from Reviewer zKFv).** Although in our experiments, the fine-tuning of diffusion models took a considerable amount of time (10 days), we gently remind the reviewer that this was done using a single NVIDIA GeForce RTX 3090 Ti GPU. The time required for this process can be significantly reduced with access to more powerful computing resources.

Given the fine-tuned diffusion model, our method achieves comparable runtime to baseline methods for training per-scene NeRF. We will soon provide an analysis of the runtime of our method and a comparison of the time taken by us and the baseline methods.

**Generalization to Other Benchmarks (Question from Reviewer beZg and Question 1 from Reviewer zKFv).** In the appendix, we report our experimental results on the LLFF dataset. Since the model was trained on the Hypersim dataset (indoor scenes) and was not fine-tuned on LLFF, an expected performance degradation was observed. However, our method still outperforms the majority of baselines. Looking ahead, our approach has the potential to generalize to more diverse scenes as large-scale datasets of general 3D scenes become more accessible in the future. Also, we have been working to validate the ability of our models trained on Hypersim to generalize to other indoor scene datasets and will report quantitative and qualitative results soon.

---

### Author Response · Authors · 2023-11-15
**Analysis on the Runtime**

In response to reviewers' common concern about the computational consumption of our approach, we discuss it here.

**Fine-Tuning Diffusion Model.** Our model was fine-tuned using a single NVIDIA GeForce RTX 3090 Ti GPU over a period of 10 days. Please kindly note that this timeframe was achieved with personal computer-level resources. Enhanced computational resources and advanced techniques [1] should significantly reduce this duration. Furthermore, this is a one-time process, allowing the adapted model to be applied directly to per-scene NeRF training without further fine-tuning.

**Per-Scene NeRF Training.** With the fine-tuned diffusion model, we conducted per-scene NeRF training on a single NVIDIA GeForce RTX 3090 Ti GPU. The purpose of the process is twofold: 1) training the radiance field and 2) generating pseudo-observations with the diffusion model. For example, in an experiment on Hypersim with 10 training views, our progressive training approach produced 40 images (50% discarded due to defects), taking 3 minutes. The bulk of the time (42 minutes) was for radiance field training, with progressive training at $N_\text{iter}=3$ resulting in $4 \times 30000$ training steps. The time spent on both aspects is summarized in the following table:

| Synthesizing Pseudo-Observations | Radiance Fields Training |
|:--------------------------------:|:------------------------:|
|            3 minutes             |       42 minutes         |

In response to the point raised by Reviewer QhhZ, our method, due to progressive training, involves a sequence of radiance field learning
processes. A comparative analysis of the time with and without progressive strategy is presented below. The times reported encompass both pseudo-observation generation and serial radiance field optimization.
| w/ progressive | w/o progressive |
|:--------------:|:---------------:|
|   45 minutes   |    22 minutes   |

Despite integrating a diffusion model and serial radiance field training, our approach does not markedly increase runtime. The diffusion model's role is limited to a mere 3 minutes for pseudo-observation synthesis. This is in contrast to recent studies that combine NeRF and diffusion models, which necessitate diffusion model inference at each NeRF training step. This efficiency significantly reduces computational overhead. In fact, our method is faster than most baselines, which generally exceed an hour. (Since various methods implement upon different frameworks in their official codes, we refrain from a formal runtime comparison, providing this only as a reference.)

**References**
[1] Hu et al., “Lora: Low-rank adaptation of large language models”.

---

### Author Response · Authors · 2023-11-23
**Updated Manuscript**

We would like to express our gratitude once again to each reviewer for their insightful feedback on our work. In response to their valuable suggestions, we conducted additional experiments and revised our manuscript. To facilitate easy identification, we highlighted all updates in blue. We hope these updates comprehensively address the reviewers' concerns and enhance the quality of our paper.
Major updates to our manuscript include:
1. **Experiments on Additional Datasets:** We included experiments on two additional datasets to showcase the generalization ability of our method. （Section E)
2. **Evaluation of Pseudo-Observations:** We added an experimental evaluation for the pseudo-observations our deceptive diffusion model synthesizes, comparing them to those from image restoration models. (Section D)
3. **Ablation Study on Filtering Strategy:** We performed a new ablation study on our filtering strategy.
4. **Discussions on Computational Cost and Generalization:** We added discussions about the computational cost and generalization ability of our method.

---

### Meta-Review · Area_Chair_FLw6 · 2023-12-04

**Metareview:**

This paper investigates NeRF reconstruction in few-shot setting. It proposes to improve reconstruction by Deceptive-NeRF, which trains a NeRF model iteratively using data augmented with photo-realistic synthetic images generated from diffusion models. Experiments show that the proposed approach outperforms other baselines using neural fields.

Strengths:
+ The problem of leveraging 2D generative models for 3D is important.
+ The idea of employing diffusion model to refine images rendered from a coarse NeRF is reasonable.
+ The performance is better than baselines.

Weaknesses:
- Some diffusion-synthesized samples are view-inconsistent.
- The evaluation on the synthetic images generated from diffusion models is insufficient.
- The progressive training is computationally expensive.
- The diffusion model is trained with limited data and has difficulty to generalize to other benchmarks.
- The paper is not well written and needs to improve.

**Justification For Why Not Higher Score:**

This is a borderline paper. Reviewers acknowledge that this investigation is worthwhile. However, there are shared concerns on the computational consumption of the approach and generalization to other benchmarks.
The authors address some of these, providing additional experiments and revised paper, but these were not enough to sway reviewers. I think this paper is not ready for publication at the current stage due to the weaknesses listed in the summary.

**Justification For Why Not Lower Score:**

N/A

---

### Decision · Program_Chairs · 2024-01-16

Reject